# Combining ablative radiotherapy and anti CD47 monoclonal antibody improves infiltration of immune cells in tumor microenvironments

Elham Rostami[1], Mohsen Bakhshandeh[2], Haniyeh Ghaffari-Nazari[1], Maedeh Alinezhad[3], Masoumeh Alimohammadi[4], Reza Alimohammadi[3], Ghanbar Mahmoodi Chalbatani[4], Ehsan Hejazi[5], Thomas J. Webster[6], Jalil Tavakkol-Afshari[1]*, Seyed Amir Jalali[3]*

1 Department of Immunology, School of Medicine, Mashhad University of Medical Sciences, Mashhad, Iran, 2 Department of Radiology Technology, Allied Medical Faculty, Shahid Beheshti University of Medical Sciences, Tehran, Iran, 3 Department of Immunology, Medical School, Shahid Beheshti University of Medical Sciences, Tehran, Iran, 4 Department of Oncology, Tumor Immunotherapy and Microenvironment Group, Luxembourg Institute of Health, Luxembourg City, Luxembourg, 5 Department of Clinical Nutrition and Dietetics, School of Nutrition Sciences and Food Technology, Shahid Beheshti University of Medical Sciences, Tehran, Iran, 6 Department of Chemical Engineering, Northeastern University, Boston, MA, United States of America

* jalalia@sbmu.ac.ir, jalali5139@yahoo.com (SAJ); Tavakolaj@mums.ac.ir (JTA)

**Data Availability Statement:** All relevant data are within the article.

## Abstract

Radiotherapy as an anti-tumor treatment can stimulate the immune system. However, irradiated tumor cells express CD47 to escape the anti-tumor immune response. Anti- CD47 Immunotherapy is a possible way to tackle this problem. This study evaluated the effect of single high dose radiotherapy combined with an anti-CD47 monoclonal antibody (αCD47 mAb) in CT26 tumor-bearing BALB/c mice. We assessed the tumors volume and survival in mice 60 days after tumor implantation. Also, immune cell changes were analyzed by flow cytometry in tumors, lymph nodes, and spleen. Combination therapy enhanced the anti-tumor response in treated mice by increasing CD8+ T cells and M1 macrophages and decreasing M2 macrophages and myeloid-derived suppressor cells (MDSCs) in the tumor microenvironment (TME). Also, our results showed that combination therapy increased survival time in mice compared to other groups. Furthermore, tumor volumes remarkably decreased in mice that received a single high dose RT plus αCD47 mAb. In conclusion, we showed that combining RT and αCD47 mAb improved the immune cell population in TME, regressed tumor growth, and increased survival in tumor-bearing mice.

## Introduction

Immunotherapy has been developed to utilize the host immune system against tumors selectively. However, immunotherapies' function is dependent on a pre-existing anti-tumor immune response. Radiotherapy (RT) is a curative or palliative treatment for many

**Funding:** 1. Mr Jalil Tavakkol-Afshari, Grant number:960553, Funder name: Mashhad University of Medical Sciences, URL: https://www.mums.ac.ir/index.php/en/ 2. Mr Seyed Amir Jalali, Grant number:97000722, Funder name: Iran National Science Foundation, URL: https://insf.org/en The funders had no role in study design, data collection and analysis, decision to publish, or preparation of the manuscript.

**Competing interests:** The authors have declared that no competing interests exist.

**Abbreviations:** RT, radiotherapy; CD47, cluster of differentiation 47; αCD47, mAb, anti-CD47 monoclonal antibody; MDSCs, myeloid-derived suppressor cells; TME, tumor microenvironment; DAMPs, damage-associated molecular patterns; ICD, induces immunological cell death; TAMs, tumor-associated macrophages; PD1, programmed cell death protein 1; PDL1, programmed cell death ligand 1; CTLA4, Cytotoxic T-Lymphocyte Associated Protein 4; TSP1, thrombospondin-1; SIRPα, signal regulatory protein-α; TILs, tumor infiltrating lymphocytes; BED, biological effect dose; RPMI media, Roswell Park Memorial Institute media; FBS, fetal bovine serum; Gy, gray; TTE, time to endpoint; TGD, tumor growth delay; FOXP3, forkhead box P3; IFNγ, Interferon gamma; DLNs, drainage lymph nodes.

malignancies. RT not only kills tumor cells directly but also can indirectly kill tumor cells by stimulating the anti-tumor immune response in both irradiated and out-of-field sites. Therefore, RT can trigger and promote the systemic immune response required for immunotherapy effectiveness [1].

RT induces irreversible DNA damage and cell cycle arrest, resulting in cell death via apoptosis, necrosis, or autophagy in tumor cells [2]. RT stimulates releasing the Damage-Associated Molecular Patterns (DAMPs) and induces immunological cell death (ICD) [3]. Irradiated tumor cells act as an in situ vaccine that activates the immune system response by activating dendritic cells, enhancing tumor antigen cross-presentation, and elevation of tumor-infiltrating lymphocytes [2, 4].

In addition, RT can negatively affect the anti-tumor immune system. Immunotherapies are promising approaches to overcome RT-mediated immune response suppression. Immunosuppressive effects of RT in the TME include an increased infiltration of regulatory T cells (Tregs), M2 tumor-associated macrophages (TAMs), and MDSCs [5], along with the expression of immune checkpoints molecules such as PD1/PDL1, CTLA4, and CD47 on tumor cells [6]. Some preclinical studies have reported radiation increased CD47 expression in TME [7, 8] and a combination of CD47 blocking and radiotherapy has synergistic effects on tumor growth regression [8–11].

CD47 is an immune checkpoint receptor widely expressed in normal tissues. CD47 surface glycoprotein belongs to the immunoglobulin superfamily that acts as a receptor for thrombospondin-1 (TSP1) and ligand for signal regulatory protein-α (SIRPα). CD47 binds to SIRPα on the macrophages, inhibits phagocytosis, induces a "don't eat me" signal to the macrophages, and acts as a self-marker for regulating phagocytosis. CD47 is overexpressed in a large number of cancers and is considered a target for cancer immunotherapy. Expressed CD47 on tumor cells can bind to SIRPα on the macrophages and inhibit phagocytosis of tumor cells [12]. Also, there is evidence regarding the role of CD47 in T cell exhaustion. Tumor-infiltrating lymphocyte (TIL) analysis revealed up-regulation of CD47 in CD8+ T cells and T cell exhaustion [13]. Although RT kills tumor cells, it can damage normal cells adjacent to the tumor, limiting RT effectiveness. CD47 Inhibition in combination with RT can protect normal cells against RT while increasing the sensitivity of tumor cells to RT and impairing tumor progression [9, 10].

Our previous study investigated the effect of different RT regimens on immune cells population, tumor volume, and survival in a CT26 tumor mice model. We utilized three different fractionation schemes; a single high-dose, hypofractionated, and hyperfractional (conventional) RT with the same biological effect dose (BED). Compared to another regimen, a single high dose RT augmented the numbers of immune cells in the TME and was the most efficient in combination with αPDL-1 mAb [14]. Therefore, we decided to examine the potency of combination therapy of single high dose RT plus anti-CD47 immune checkpoint inhibitor compared to single therapies for tumor control. Here, we conducted a study to compare immune responses resulting from the combination of radiotherapy and immunotherapy with single therapies and assayed the effects of different treatments on the tumor regression and survival of CT26 tumor-bearing BALB/c mice.

## Material and methods

### Mice and tumor challenge

Female BALB/c mice (6 to 8 week old) were obtained from the Royan Institute of Iran. The animals were housed under standard conditions with a 12h/12h light-dark cycle with constant temperature and humidity and free access to food and water. This study was carried out in strict accordance with the recommendations in the Helsinki declaration. The protocol was

approved by the Institutional Ethical Committee and Research Advisory Committee of Mashhad University of Medical Sciences (Ethic number code: IR.MUMS.fm.REC.1396.495). Murine CT26 colon carcinoma cells were obtained from the Pasteur Institute of Iran. These cells were cultured in RPMI-1640 media (Gibco, US), supplied with 10% heat-inactivated fetal bovine serum (FBS) (Cegrogen, Germany), 10,000 IU/ml penicillin, and 10,000 mg/ml streptomycin in 5% $CO_2$ at 37° C. The viability of the cells was assessed by a trypan blue dye exclusion before the experiment. BALB/c mice were anesthetized before tumor challenge to reduce pain or suffering. 12.5 mg/kg Xylazine 2% and 100 mg/kg ketamine 10% injected intraperitoneally into mice (Alfasan, Sofia, Bulgaria). Then they were injected subcutaneously with $1\times10^6$ CT26 cells into the right flank to induce a tumor.

### Evaluation CD47 expression after ablative RT

Before the main experiments, six mice were selected for the evaluation of CD47 expression after a single high dose (ablative) RT. Ablative RT was performed on 3 mice according to the following treatment protocol and 3 mice were considered as controls. Mice were sacrificed seven days after RT via cervical dislocation. Tumor tissue from each mouse was removed and the expression of CD47 on TILs was determined by flow cytometry for the Alexa Fluor 488 anti-mouse CD47 Antibody (Biolegend, San Diego, California) according to the following protocol.

### Treatment

Tumor-bearing mice were randomly distributed into four groups and eight mice in each group. 1. RT group that received ablative radiation (16Gy ×1F). 2. αCD47 group that received anti-mouse CD47 monoclonal antibody. 3. RT+ αCD47 mAb group that received the combination of both ablative radiation and anti-mouse CD47 mAb. 4. untreated group (control).

Irradiations were performed 16 days after tumor inoculation (when the tumor volume was reached at least 200–300 mm³). For this purpose, a clinical linear accelerator (6 MV photons, Elekta synergy linear accelerator, Stockholm, SE) was used. To reduce pain or suffering the mice were anesthetized before irradiation with Xylazine and ketamine intraperitoneally. The modified 50 ml plastic tube was considered for mice radiotherapy. The tumor area in the tube was exposed for irradiation. The rest of the body was protected with a lead plate of 9 cm thickness. Ablative RT was administered in a single fraction of 16 gray (Gy) locally to the tumors in the right flank. A field of 3×3 cm² and 5 mm margins on each side were considered for RT. Beam delivery was completed 350 Gy/min tangentially with a 6 MV X-ray. A super flab bolus material of 1.5 cm was located above the tumor. The distance between the source and mouse skin was 100 cm [14, 15].

The αCD47 mAb: InVivoMAb anti-mouse CD47 (IAP), 9H10 clone (Bio X Cell, NH, USA) was administered intraperitoneally on day 17 (one day after radiotherapy), day 19, and day 21 (Fig 2A).

### Tumor volume and survival

Tumor volume was evaluated two times a week. Tumor size was measured with a digital caliper and tumor volumes were calculated by $(a \times b^2)/2$ formula (a, length; b, width). The endpoint criteria in mice were tumor growth greater than 1500 mm³, bodyweight decreases by more than 15% of the initial weight, and health decline, death, or visible symptoms of sickness. Time to Endpoint (TTE) and percent of Tumor Growth Delay (%TGD) was also calculated in the controls and treated groups.

TTE = [log (endpoint)-b]/m. TTE diagram is driven by the linear regression of the log of tumor growth at times. (b) is the y-intercept and (m) is the slope.

%TGD = (T − C) / C × 100, (T) is the mean TTE of the treatment group, and (C) is the mean TTE of the control group

## Flow cytometry analysis

Flow cytometry analysis was performed on the TILs, spleen, and drainage lymph nodes (DLNs) using a BD FACSCalibur Flow Cytometer; all Flow Cytometry Antibodies purchased from Biolegend, San Diego, California.

After 23 days of tumor induction, three mice per group were sacrificed via cervical dislocation. Tumor tissues from mice were removed and diced into small pieces. Digestion of the tumor for single-cell separation was completed in RPMI 1640 containing collagenase type I (2mg\mL) (Thermo scientific) and DNAse type I (10 U/mL) (Gibco) at 37˚C for one hour. The cells were then filtered through a 70 μm cell strainer (BD Falcon, USA) and centrifuged at 1500 rpm for 10 min.

The spleen was removed, minced into small pieces, and the pellet washed with RBC lysis buffer (cyto matin gene, Iran) for 5 min, and dispersed in 5ml of RPMI-1640 medium. Lymph nodes near the tumor also were removed, mined, and dispersed in 5ml of RPMI-1640 medium. The cells then were filtered through a 70 μm cell strainer (BD Falcon, USA) and centrifuged at 1500 rpm for 10 min. The pellet of cells was resuspended in the flow cytometry staining buffer.

The TILs, spleen, and DLNs cells were first stained with Zombie NIR for live/dead discrimination, and then flow cytometry analyzed using APC anti-mouse CD45, PerCP anti-mouse CD4, PE anti-mouse CD8, PE anti-mouse CD25, and an Alexa Fluor 488 anti-mouse FOXP3 antibody. Flow cytometry analyses of TILs were also completed using the PerCP anti-mouse CD11b, Alexa Fluor 488 anti-mouse CD80, FITC anti-mouse Gr-1, and PE anti-mouse CD206 antibody. Foxp3 intracellular staining was performed according to the Biolegend True-Nuclear™ transcription factor manufacturer's instructions. First, surface antigens were stained and fixed. Then cells were permed with fixation/permeabilization buffer and stained by anti-mouse -FOXP3 antibody for 30 minutes at room temperature in the dark. For evaluation of immune cell changes in the Tumor, spleen, and lymph nodes, we selected live cells and gated CD45+ cells. Then, we gated specific cells and compared the percentage of different immune cells in the control groups with treatment groups according to the specific markers.

## Measurement of IFNγ production in the spleen

A cell activation cocktail was used for spleen cell cultures. This cocktail contained PMA/Ionomycin with Brefeldin A (Biolegend, San Diego, California). It was added to the spleen cells and incubated for 4 hours, then centrifuged at 300g 10 min. The pellet was then suspended in a flow cytometry staining buffer and stained using FITC fluorochrome antibodies for IFNγ (Biolegend, San Diego, California).

## Statistical analysis

FlowJo 10.6.2 software was used for flow cytometry data analysis. This software was used to evaluate the cells stained with flow cytometry antibodies and to determine the percentage of these cells in different tissues. Also this software were used for evaluation of CD47, Foxp3, and IFNγ expression in cells and calculation of mean fluorescence intensity (MFI). Then, the final flow cytometry data were analyzed with GraphPad Prism version 8. Averages are presented as the mean ± the standard deviation (SD). Descriptive statistics, independent t-tests, and post

hoc tests for one-way ANOVA were utilized. Log-rank Mantel-Cox tests were used for the Analyses of Survival. P values < 0.05 were considered statistically significant.

## Results

### Increased CD47 expression in tumor tissue following ablative RT

To determine the effect of ablative RT on CD47 expression, six mice with tumors were selected. Ablative RT was performed on 3 mice and 3 mice were considered as controls. Seven days after RT, CD45- CD47+ live cell was evaluated. It was observed that the percent of these cells increased and CD47 MFI in tumor cells also increased significantly due to ablative RT (p < 0.05) (Fig 1).

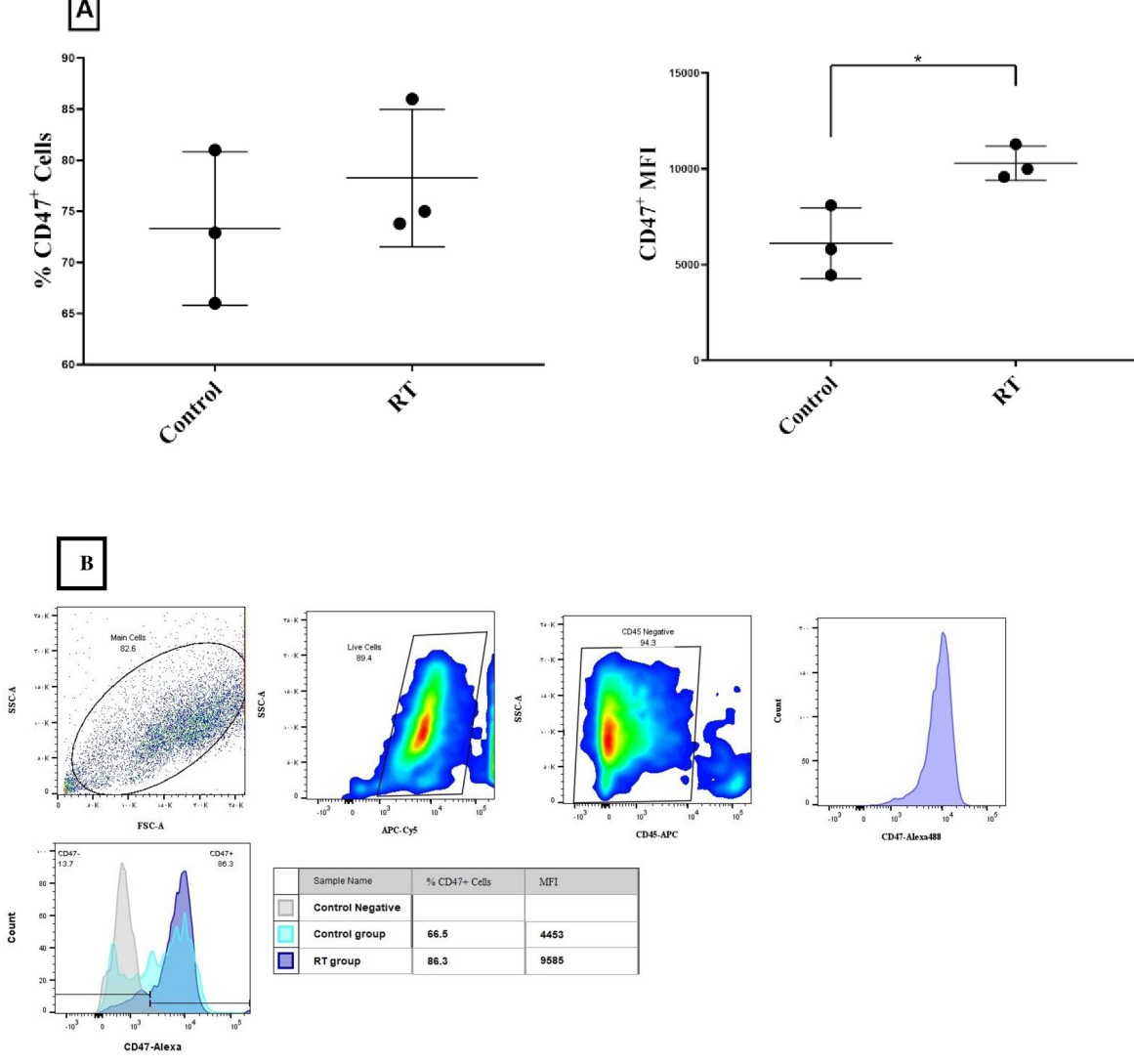

**Fig 1. Ablative RT increased CD47 expression in tumors.** $1 \times 10^6$ CT26 cells were injected subcutaneously into the right flanks to induce the tumor. Tumor induction day is the first day of the study. RT was performed on day 16 (1×16Gy). On day 23, 6 mice were sacrificed and investigated for CD47 expression by flow cytometry. Data are displayed as the mean (SD) and analyzed using unpaired T Test (*: P < 0.05) (A). Gating scheme used in TCD47+ phenotyping by flow cytometry in tumor tissue and relevant fluorescence images of these cells in studied groups (B).

## Ablative RT combined with αCD47 mAb increased T CD8$^+$ cells infiltration into tumors

The combination therapy plans are shown in Fig 2A. The percentage of CD45$^+$CD8$^+$ live T cells in the tumor was significantly greater in mice from combining RT and the αCD47group than those from the control group ($p<0.01$) and the αCD47 group alone ($p<0.01$) but not compared to RT alone group. The percentage of CD8$^+$ T cells in the RT group was significantly greater than the control group ($p <0.05$) and the αCD47 group ($p<0.05$) (Figs 2B and 4A).

The percentage of CD45$^+$ CD8$^+$ live T cells in the spleen and drainage lymph nodes was not significantly different between the groups (Fig 2B). The percentage of CD45$^+$CD4$^+$ CD25$^+$ FOXp3$^+$ live T cells was significantly less in the drainage lymph nodes in mice after combining the RT and αCD47 group than in those from the control group ($p<0.0001$) and the αCD47 group ($p < 0.001$) but not compared to the RT group (Fig 2D). Also, FOXP3 MFI was significantly less in the drainage lymph nodes in mice from the combining RT and αCD47group than the control group ($p<0.05$) but not compared to the αCD47 and RT group (Figs 2E and 4B).

## Ablative RT combined with αCD47 mAb increased the numbers of M1 macrophages and decreased the numbers of M2 macrophages and MDSC cells in TME

The percentage of M1 macrophages cells (CD45$^+$CD11b$^+$CD80$^+$) was greater in mice from combining RT and the αCD47group than in those from the control group ($p < 0.05$), αCD47 group ($p < 0.05$) and RT group (non-significant) (Figs 3A and 4C). The percentage of M2 macrophages (CD45$^+$CD11b$^+$CD206$^+$) in the combined RT and αCD47group was less than the control group ($p < 0.05$), αCD47 group ($p< 0.05$) and RT group (not significant) (Figs 3B and 4C). The M1/ M2 ratio was greater in mice from the combined RT and αCD47group than in those from the control group ($p < 0.01$), αCD47 group ($p < 0.001$) and RT group ($p < 0.01$) (Fig 3C).

The percentage of MDSC cells (CD45$^+$CD11b$^+$Gr-1$^+$) in the combined RT and αCD47 group was significantly less than the control group ($p < 0.0001$), αCD47 group ($p < 0.0001$), and RT group ($p < 0.001$). Also, the percentage of MDSC cells in the RT group was significantly less than the control group ($p < 0.01$) and αCD47 group ($p < 0.01$) (Figs 3D and 4C).

## Ablative RT combined with αCD47mAb increased IFNγ expressing cells in the spleen

Effector cytokine IFNγ was analyzed in the spleen. It was found that for the combined ablative radiation and αCD47 mAb group, CD45$^+$CD8$^+$ immune cells expressing IFNγ increased compared to the control group ($p < 0.01$) and the RT group ($p < 0.05$). These cells also increased in the αCD47 group compared to the control group ($p < 0.05$) (Fig 5).

## Tumor size for the combined RT and αCD47mAb significantly decreased 60 days after tumor implantation

The therapeutic effect of αCD47 combined with RT was evaluated by tumor size assessment and survival in tumor-bearing mice. Tumor growth for the different mice in each treatment group 60 days after induction of reached the endpoint before day 45. The RT on average only slowed the growth of the tumor but combining RT and αCD47 mAb reduced tumor size for up to 45 days (Fig 6B). In combining the RT and αCD47 group and RT group, tumor size on

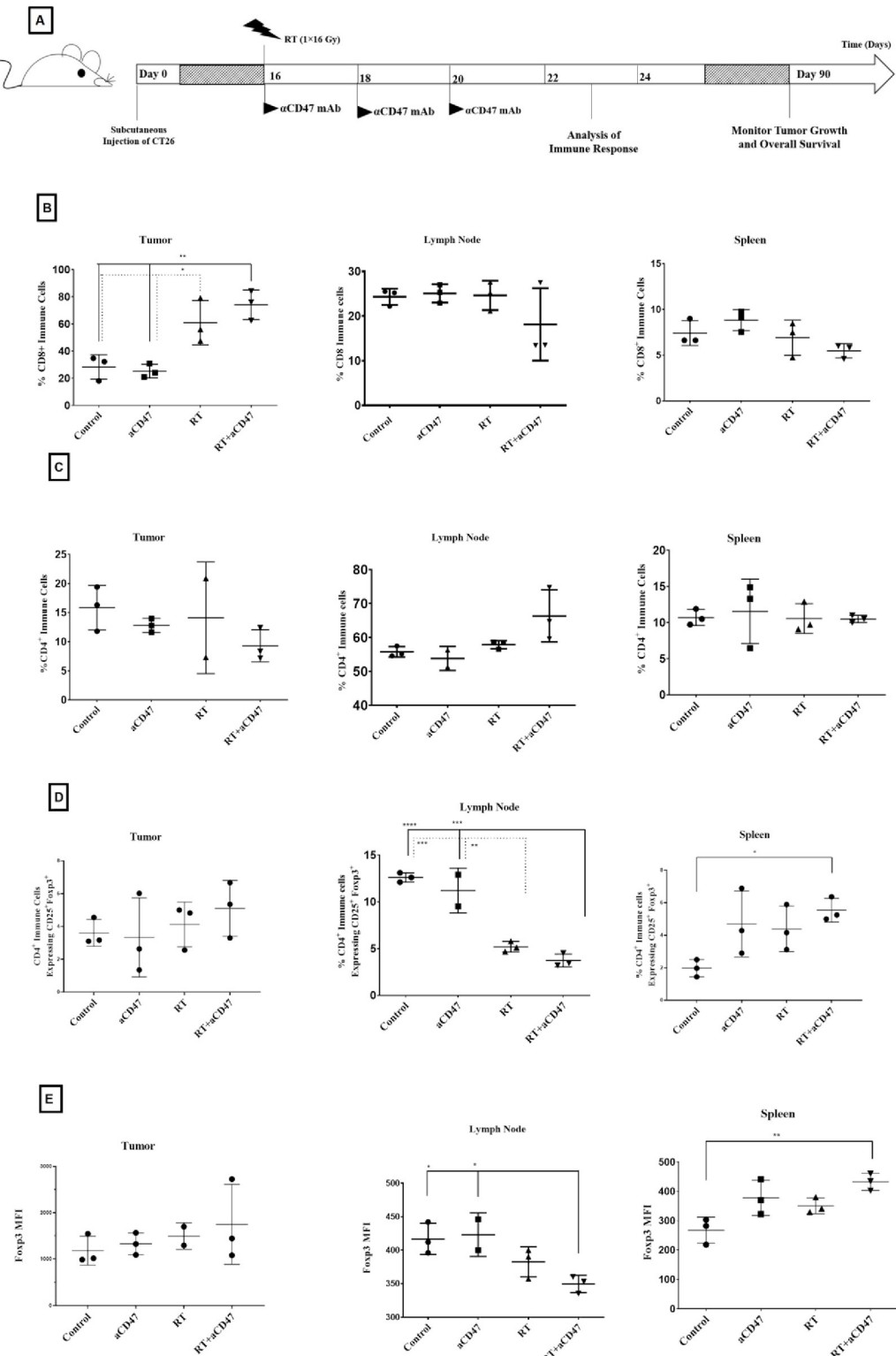

**Fig 2. CD8+ cell infiltration increased due to combination of ablative RT and αCD47 mAb in tumors.** The schematic plan of combining ablative RT with αCD47 mAb is shown, combining ablative RT with αCD47 mAb (n = 9), RT (n = 9), αCD47 (n = 8), and the control (n = 9). Tumor induction was completed with $1 \times 10^6$ CT26 cells inoculated into the right flanks of the mice subcutaneously. Tumor induction day as the first day of study. On day 16, RT was performed (1×16Gy). On day 23, 3 mice of each group were analyzed by flow cytometry. In combining RT and αCD47 mAb groups,

mice received 100 µg/ml of αCD47 mAb intraperitoneally on days 16, 18, and 20 with ablative RT (1×16Gy) on day 16 (A). The percentage of live CD45+cells that are CD4+, CD8+, and CD4+CD25+Foxp3+and infiltrated into the spleen, lymph node, and tumor are shown in (B), (C), and (D) respectively. Foxp3 MFI in the different groups are presented in (E). Data are displayed as the mean (SD) and were analyzed using the Tukey's Multiple Comparison Test (*: P < 0.05, **: P < 0.01, ***: P < 0.001, and ****: P < 0.0001).

day 23 after tumor inoculation was significantly less compared to the control (p < 0.0001) and αCD47 groups (p < 0.0001). 45 days after tumor inoculation, the tumor size regression in the combined RT and αCD47 group was significant compared to the control (p < 0.001), αCD47 (p < 0.001), and RT groups (p < 0.05). 60 days after tumor inoculation, the tumor size in the combined RT and αCD47 group was less compared to the control and the αCD47 only group (p < 0.05) (Fig 6C). Lastly, the survival rate after 60 days was the greatest in the combined RT and αCD47 group (83.33%) followed by the RT group (33.33%). No mice from the αCD47 and control groups survived to day 60, but 4 mice in the combined RT and αCD47 groups and 2 mice in the RT groups possessed completely regressed tumors (Fig 6D).

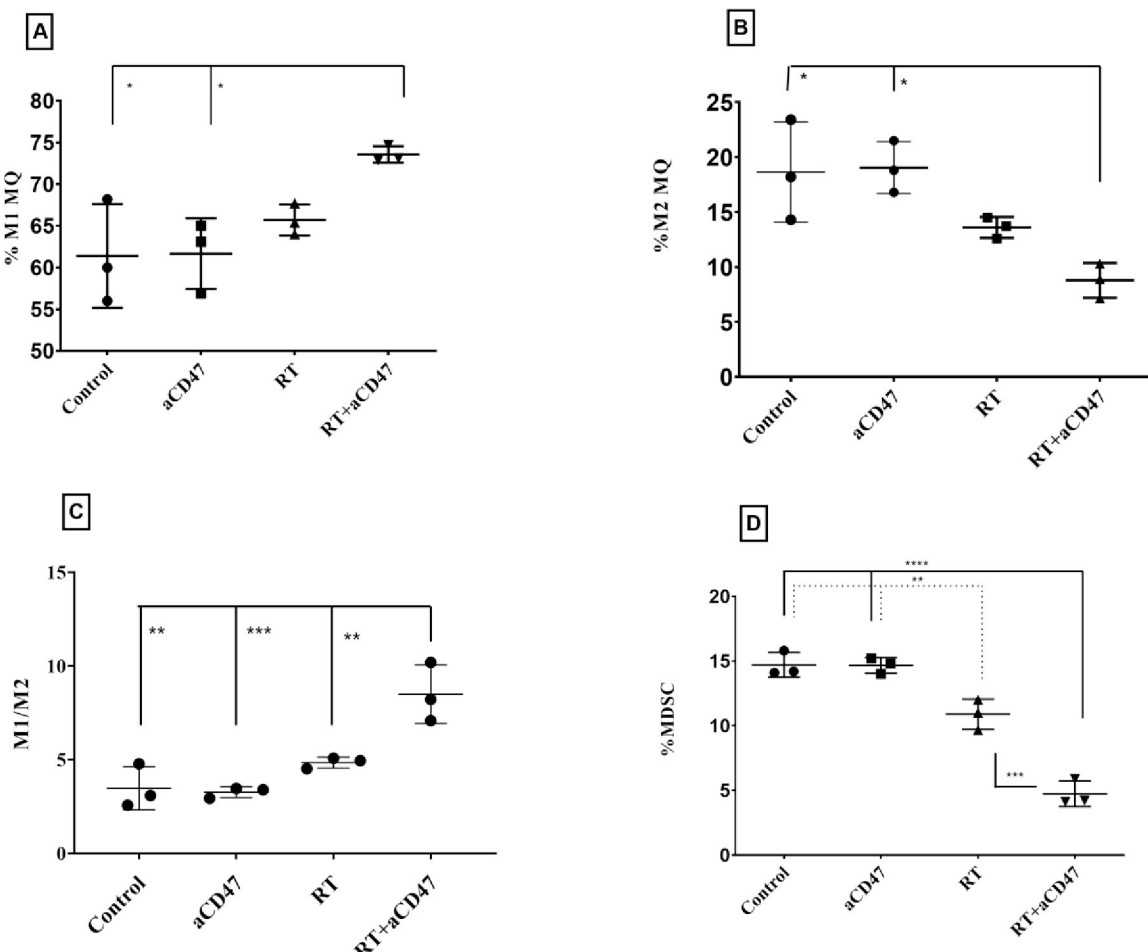

**Fig 3. Ablative RT combined with αCD47 mAb altered the numbers of M1, M2 macrophages, and MDSC cells in TME.** The percentage of CD45+CD11b+CD80+ (M1 macrophages), CD45+CD11b+CD206+ (M2 macrophages), and live CD45+CD11b+Gr-1+ (MDSC) cells infiltrating into the tumor are presented respectively in (**A**), (**B**), and (**D**). The M1/M2 macrophage ratio is presented in (**C**) Data are displayed as the mean (SD) and were analyzed using the Tukey's Multiple Comparison Test (*: P <0.05, **: P <0.01, ***: P <0.001, and ****: P <0.0001).

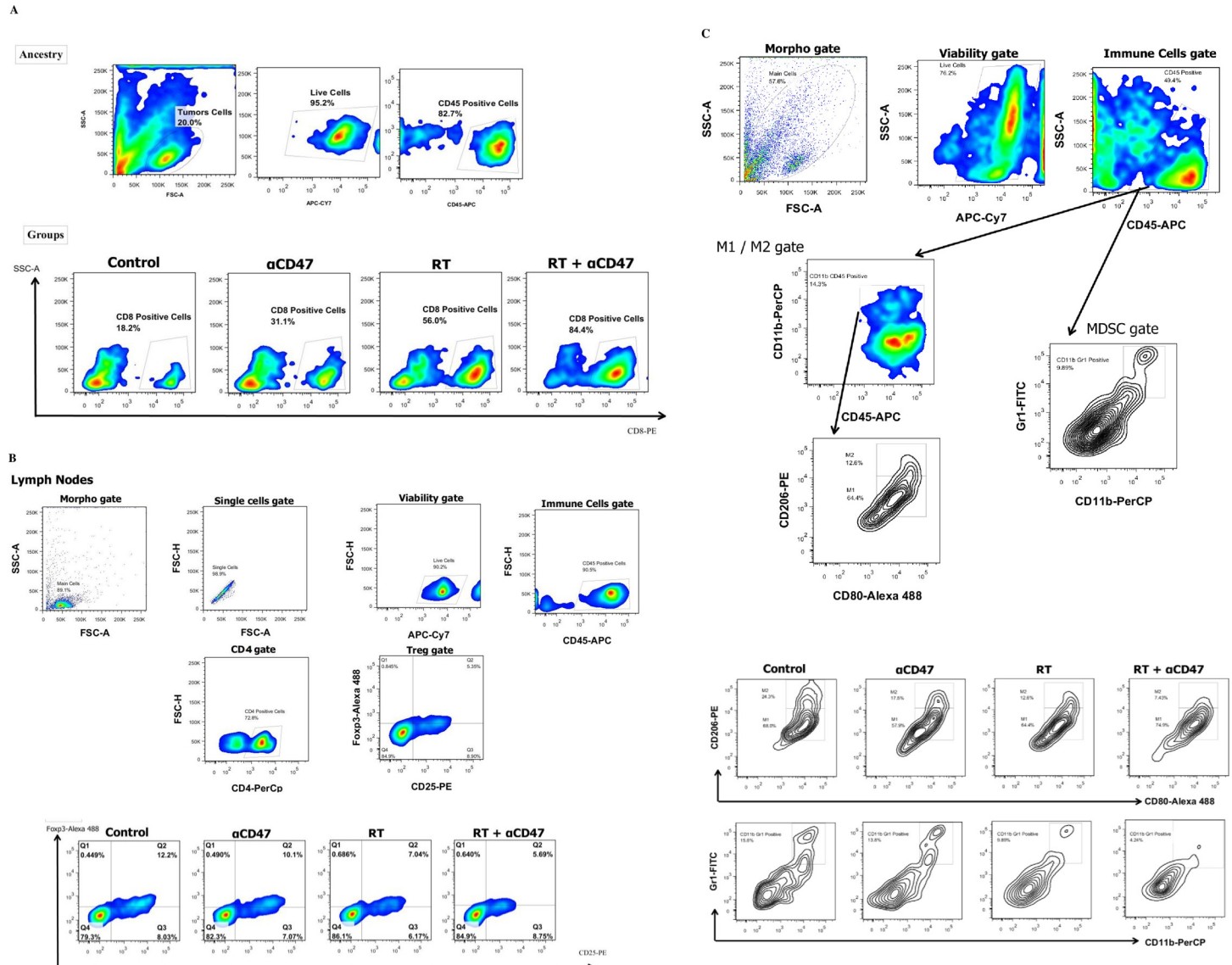

**Fig 4. Fluorescence images for identification of CD8+, Treg, M1, M2 macrophages, MDSC cells.** Live/CD45 positive populations were gated as immune cells using FlowJo (version 10.6.2). Gating scheme used in TCD8+ phenotyping by flow cytometry in tumor tissue and relevant fluorescence images of these cells in studied groups (A). Gating scheme used in Treg phenotyping by flow cytometry in DLNs and relevant fluorescence images of these cells in studied groups (B). Gating scheme used in M1, M2 macrophages, and MDSC phenotyping by flow cytometry in tumor tissue and relevant fluorescence images of these cells in studied groups (C).

## TTE and TGD in the combined RT and αCD47 group were greater than the other groups

TTE of mice that received the combined RT and αCD47 were greater than the RT group and followed the order of group αCD47 and the control. The tumor growth delay (TGD) was also the greatest in combining RT and the αCD47 group and followed the order of groups RT and αCD47 90 days after the start of the study. Lastly, tumor growth delay from the combined RT and αCD47 group was 80.58% greater than with the RT group (Table 1).

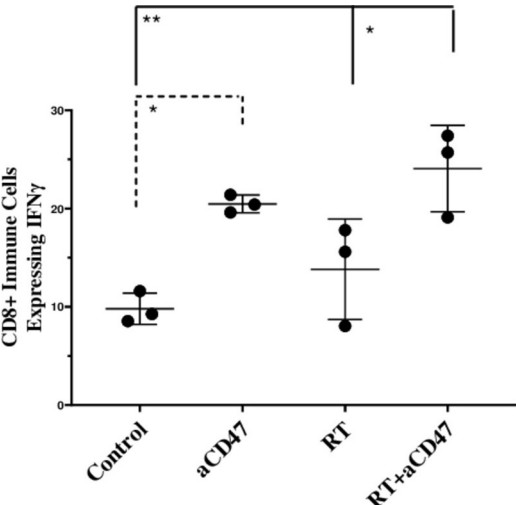

**Fig 5. Ablative RT combined with αCD47mAb increased IFNγ expressing cells in tumors.** The percentage of CD8⁺cells that expressed IFNγ in the spleen are shown. Data are displayed as the mean (SD) and were analyzed using the Tukey's Multiple Comparison Test (*: P <0.05, **: P <0.01).

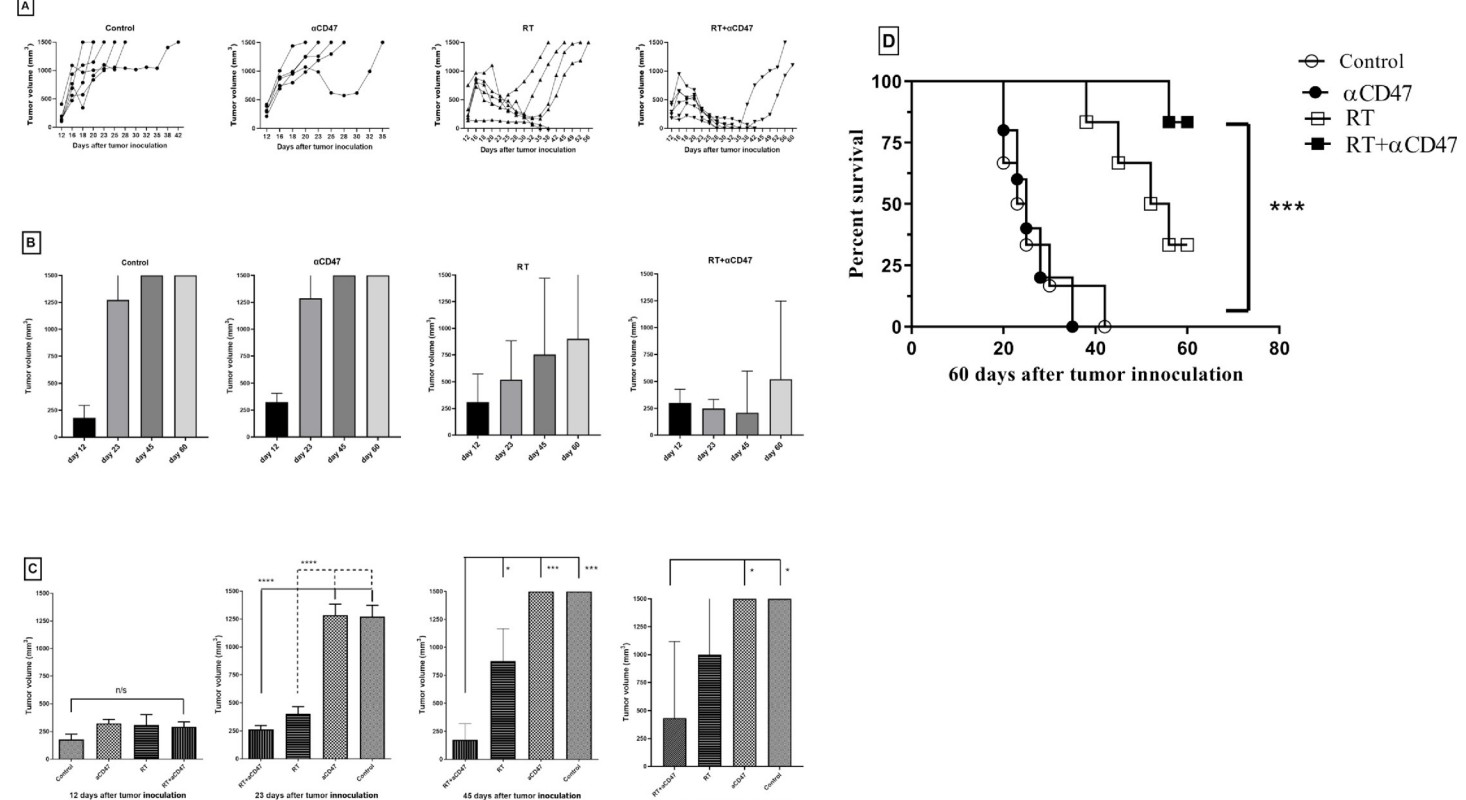

**Fig 6. Combining RT and αCD47 reduced tumor volume and increased survival rates.** The tumor size change for all the groups during the study is shown (A). The average tumor volume for each group at 12, 23, 45, and 60 days after induction of the tumor is shown (B). The differences between the average tumor volume in groups at 12, 23, 45, and 60 days after induction of the tumor are shown. Tumor volumes averaged at the start of the study 100–300 mm³ and there was no significant difference between groups (C). Data are displayed as means (SD) and were analyzed using the Tukey's Multiple Comparison Test (*: P < 0.05, ***: P < 0.001, and ****: P < 0.0001). Survival of the tumor-bearing mice is shown in (D). Survival data were analyzed using the Log-rank (Mantel-Cox) Test (***: P < 0.001).

**Table 1. The time to endpoint (TTE) and tumor growth delay (%TGD) between the combined RT and αCD47, RT, αCD47, and control groups were calculated.**

| Groups | Average of TTE (Day) | % TGD Compared to Control Group | % TGD Compared to αCD47 Group |
|---|---|---|---|
| RT+ αCD47 | 79/84 | 223/5 | 216/70 |
| RT | 60/39 | 142/92 | 139/55 |
| αCD47 | 25/21 | 2/15 | 0 |
| Control | 24/68 | 0 | -2/10 |

## Discussion

In this study, we evaluated the effect of combining ablative RT and αCD47mAb on the immune response, tumor size, and survival in a CT26 tumor model of BALB/c mice. Our data show that the combined RT and αCD47 mAb increased infiltration of TCD8[+] and M1 macrophages and decreased M2 macrophages and MDSC cells in the TME significantly. Tumor size in the combined group was significantly less than the control, αCD47, and RT groups after 45 days of tumor implantation. Also, the survival rate in the combined group was the greatest during the study.

Evaluation of changes in immune cells population due to combination of ablative RT and αCD47 mAb has not been studied, which represents a key novel contribution of the present study. The present results indicated that the number of CD8[+] cells in the TME increased significantly after the combined ablative RT and αCD47 mAb treatment. Our previous study, Alinezhad *et al.*, showed that ablative RT can increase the TCD8[+] population in TME [14]. Also, other studies revealed that ablative RT can improve TCD8[+]cell infiltration to the TME [16–18]. Tumor-infiltrating CD8[+] T cell numbers are a critical prognostic factor for cancer [19, 20]. There is a relationship between the increase of CD8[+] T cells in TME with a decrease in tumor volume [21] and an increase in survival [20]. Thus, the increase in CD8[+] cells can be related to a reduction in tumor volume and improvement of animal survival in our study. Due to this, it is suggested that the following next immune mechanism related to CD8[+] T cells in TME be investigated.

In our study, the percentage of TCD8[+]INFɣ[+] cells in the spleen significantly increased in the combined group and αCD47 monotherapy group. Tao *et al.* showed that αCD47 mAb increased the level of pro-inflammatory cytokines, including IFNɣ [22]. In our previous study (Alinezhad *et al.*), we also reported that ablative RT increased TCD8[+]INFɣ[+] cells in TME [14]. M1 macrophages and CTL cells are two stimulatory cells in TME, and IFNɣ release into the TME controls their presence [23]. CTLs within the tumor stroma change macrophage polarity toward the antitumor M1 cells through paracrine signals and a reduction in the number of M2 cells [24]. Our findings showed a significantly higher number of M1 macrophages and less number of M2 macrophages in the tumor in the combined group and the M1/M2 ratio also increased. Iribarren *et al.* showed that αCD47 mAb increased M1 and decreased M2 macrophages in TME and increased the M1/M2 ratio [25] Az. Zhang *et al.* also revealed a significant increase of M1 macrophages within the tumor and shifted the phenotype of macrophages towards the M1 subtype with αCD47 mAb [26]. However, a high-dose RT can play a pro-tumorigenic effect on the macrophages population and shift macrophages toward an M2-like phenotype [27–29]. Our study supports the role of αCD47 mAb in the increase of M1 macrophages that illustrates the efficacy of the immune system's response to combination therapy.

Further, MDSC levels act as prognostic indicators of disease outcome [30]. Our results demonstrated that the percentage of MDSC cells in the combining group was significantly less than in other groups. Studies showed that a high dose of ablative RT reduced the levels of MDSC in TME while a fractionated low dose RT increased MDSC [30]. In our study, the

combined ablative RT and αCD47 more strongly decreased such events so that the MDSC level in the combined group was less than the RT group significantly and this showed a stronger decrease in the inhibitory milieu.

In this study, tumor size and survival rate were evaluated for 60 days in different mice groups. Tumor size in the combined RT (16 Gy) and αCD47 (mouse anti-CD47 Ab) was significantly less compared to the control, αCD47, and RT groups 45 days after the start of the study. In two studies, antisense suppression of CD47 using a morpholino oligonucleotide combined with RT (10 Gy) in mice, resulted in less tumor volume compared to the control group [9, 10]. Gholamin *et al.* showed in their study that the tumor growth in mice when receiving a humanized anti-CD47Ab (Hu5F9-G4) with irradiation (10 Gy), was significantly less than when receiving anαCD47 mAb or irradiation alone [11]. Also, Wang *et al.* used a CD47/SIRPα blocking peptide combined with irradiation (20 Gy) and showed that mice tumor growth was synergistically inhibited [31]. The results of our study about tumor size are similar to the studies mentioned above. The difference between these studies is in radiation therapy doses and CD47 inhibition methods.

The tumor size 60 days after tumor implantation for the combined RT and αCD47 groups was less than the RT group but not significantly. Also, after 40 days, tumor growth increased again in two mice in the combined group and they died about 70 days after tumor implantation. This may indicate that the effect of the antibody disappeared, thus, requiring antibody re-administration or the use of both antibodies and radiotherapy.

In our study, the survival analysis 60 days after tumor implantation revealed that the combined RT and αCD47 mAb significantly increased survival compared to the αCD47, control, and RT groups. Gholamin *et al.* showed that the survival rate in combining αCD47 and RT was significantly greater than RT alone, αCD47, and control [11]; these data were favorable with our results.

In conclusion, combining ablative RT and αCD47 mAb demonstrated beneficial effects on the immune response, tumor size, and survival in a CT26 model of BALB/c Mice. This combination significantly increased infiltration of functional cells and decreased suppressive cells in TME, regressed tumor growth, and improved survival rate. In addition, no specific toxicity was observed in the use of this combination therapy. No clinical signs were observed apart from tumor growth. No significant weight loss or change in the appearance of the mice was not recorded in any of the animals. In future studies, the αCD47 mAb antibody re-administration after day 45 or use 2 cycles of combination treatment composed of both RT and αCD47 can be used for more continuity in the immune response. Also, the percentage of TCD8$^+$INFγ$^+$ increased in combining the RT and αCD47 that which would trigger PDL1 expression and exhaustion of CD8$^+$ cells [32, 33] that we recommend in future studies combine ablative RT with dual mAb αCD47 and αPDL1 to improve the immune system response and overcome T cell exhaustion.

## Supporting information

**S1 Checklist. PLOS ONE humane endpoints checklist.**
(DOC)

## Acknowledgments

This article has been extracted from the thesis written by Elham Rostami in the School of Medicine, Mashhad University of Medical Sciences. (Registration No: 960553).

## Author Contributions

**Conceptualization:** Mohsen Bakhshandeh, Jalil Tavakkol-Afshari, Seyed Amir Jalali.

**Data curation:** Elham Rostami, Seyed Amir Jalali.

**Formal analysis:** Maedeh Alinezhad, Ghanbar Mahmoodi Chalbatani.

**Funding acquisition:** Jalil Tavakkol-Afshari, Seyed Amir Jalali.

**Investigation:** Elham Rostami, Haniyeh Ghaffari-Nazari, Masoumeh Alimohammadi, Reza Alimohammadi, Seyed Amir Jalali.

**Methodology:** Mohsen Bakhshandeh, Ehsan Hejazi, Seyed Amir Jalali.

**Project administration:** Elham Rostami, Seyed Amir Jalali.

**Resources:** Jalil Tavakkol-Afshari, Seyed Amir Jalali.

**Software:** Elham Rostami, Haniyeh Ghaffari-Nazari, Maedeh Alinezhad.

**Supervision:** Jalil Tavakkol-Afshari, Seyed Amir Jalali.

**Validation:** Jalil Tavakkol-Afshari, Seyed Amir Jalali.

**Visualization:** Jalil Tavakkol-Afshari, Seyed Amir Jalali.

**Writing – original draft:** Elham Rostami.

**Writing – review & editing:** Ghanbar Mahmoodi Chalbatani, Thomas J. Webster.

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
