## [Decision Letter · Decision Letter 0]

11 Mar 2022

PONE-D-22-01036Combining Ablative Radiotherapy and Anti CD47 Monoclonal Antibody Improves Infiltration of Immune Cells in Tumor MicroenvironmentsPLOS ONE

Dear Dr. Jalali,

Thank you for submitting your manuscript to PLOS ONE. After careful consideration, we feel that it has merit but does not fully meet PLOS ONE’s publication criteria as it currently stands. Therefore, we invite you to submit a revised version of the manuscript that addresses the points raised during the review process.

We look forward to receiving your revised manuscript.

Kind regards,

Jian Jian Li, M.D., Ph.D.

Academic Editor

PLOS ONE

Journal Requirements:

2. To comply with PLOS ONE submissions requirements, in your Methods section, please provide additional information on the animal research and ensure you have included details on (1) methods of sacrifice, (2) methods of anesthesia and/or analgesia, (3) efforts to alleviate suffering, (4) basic housing and health monitoring."

3. As part of your revision, please complete and submit a copy of the Full ARRIVE 2.0 Guidelines checklist, a document that aims to improve experimental reporting and reproducibility of animal studies for purposes of post-publication data analysis and reproducibility: https://arriveguidelines.org/sites/arrive/files/Author%20Checklist%20-%20Full.pdf (PDF). Please include your completed checklist as a Supporting Information file. Note that if your paper is accepted for publication, this checklist will be published as part of your article.

5. We note you have included a table to which you do not refer in the text of your manuscript. Please ensure that you refer to Table 1 in your text; if accepted, production will need this reference to link the reader to the Table.

Additional Editor Comments:

Although radiation combined with anti-CD47 immunotherapy has been reported, this work demonstrates some interesting results in mouse cancer model treated by radiation and anti-CD47 immunotherapy. However, several major concerns are raised by both reviewers which should be carefully addressed with additional data supports.

Reviewers' comments:

Reviewer's Responses to Questions

**Comments to the Author**

1. Is the manuscript technically sound, and do the data support the conclusions?

Reviewer #1: Yes

Reviewer #2: No

2. Has the statistical analysis been performed appropriately and rigorously? 

Reviewer #1: I Don't Know

Reviewer #2: Yes

3. Have the authors made all data underlying the findings in their manuscript fully available?

Reviewer #1: Yes

Reviewer #2: Yes

4. Is the manuscript presented in an intelligible fashion and written in standard English?

Reviewer #1: Yes

Reviewer #2: Yes

5. Review Comments to the Author

Reviewer #1: This study displayed some experiment results that, in comparison with radiotherapy alone, the single high dose of 16 Gy radiotherapy combined with αCD47 enhanced CT26 tumor growth delay and thus increased the survival of tumor-bearing mice, probably by increasing CD8+ T cells and M1 macrophages and decreasing M2 macrophages and myeloid-derived suppressor cells (MDSCs) in the tumor microenvironment (TME) and increasing CD8+INF� in spleen. However, these phenomena have been reported by other literatures and there is no novel mechanistic investigation in this work, although these data may helpful to strengthen the experimental evidence for tumor radioimmunotherapy.

Comments:

1. Many similar studies applied male mice as an animal model. But this work applied a female mice model. Please explain.

2. The full names of abbreviations, TILs, DLNs, TTE, TGD and so on, should be given.

3. The calculation methods of TTE, TGD should be given.

4. Fig. 1B showed the value of CD47 MFI in tumor cells. The representative fluorescence images of this MFI should be given.

5. Fig. 2A was missing in the text.

6. Corresponding to the MFI in Fig. 2E, the representative tissue images should be given.

7. Fig. 4 showed that the percentage of CD45+CD8+ immune cells expressing IFNγ was increased in spleen of both �CD47 and RT+�CD47 groups. How about it in tumors? If this increase contributes to the anti-tumor effect, why didn’t �CD47 reduce tumor growth in Fig. 5? By the way, lacking of “%” in the Y-axis of Fig. 4.

8. In Fig. 5, the tumor sizes of control and �CD47 groups approached to a maximum value 1500 mm3 on day 45 and day 60. Why had they the same volume (1500 mm3) on these different days?

9. This reviewer suggest to perform more mechanistic studies, for example, to demonstrate how M1-macrophage contributes to the anti-tumor effect of RT+�CD47 treatment.

Reviewer #2: This study evaluated the effect of a single high dose radiotherapy combined with an anti-CD47 monoclonal

antibody (αCD47 mAb) in CT26 tumor‐bearing BALB/c mice. Also, immune cell changes were analyzed by flow cytometry in tumors, lymph nodes, and spleen. Combination therapy enhanced the anti-tumor response in treated mice by increasing CD8+ T cells and M1 macrophages and decreasing M2 macrophages and myeloidderived suppressor cells (MDSCs) in the tumor microenvironment (TME).

(1) The combination of Radiotherapy and CD47 blockade was investigated before. This work did not provide novel point and evidence. The novelty is poor.

(2) Most the data were presented based on three mice in each group, which is not enough. All the data were statistical results but without original picture of flow cytometry, which greatly impaired the scientific level of the work.

(3) All the data were based on one time mice experiment but without other experiments to verify the hypothesis.

(4) Figure 2 should be combined with Figure 5, and Fig 5B-5C should be deleted. Figure 4 should be combined with Figure 1.

6. PLOS authors have the option to publish the peer review history of their article (what does this mean?). If published, this will include your full peer review and any attached files.

Reviewer #1: **Yes: **Chunlin Shao

Reviewer #2: No

---

## [Author Response · Author response to Decision Letter 0]

19 May 2022

We have revised the manuscript accordingly and provide specific answers. Response to reviewer file is attached.

---

## [Decision Letter · Decision Letter 1]

19 Jul 2022

PONE-D-22-01036R1Combining Ablative Radiotherapy and Anti CD47 Monoclonal Antibody Improves Infiltration of Immune Cells in Tumor MicroenvironmentsPLOS ONE

Dear Dr. Seyed Amir Jalali,

Thank you for submitting your manuscript to PLOS ONE. After careful consideration, we feel that it has merit but does not fully meet PLOS ONE’s publication criteria as it currently stands. Therefore, we invite you to submit a revised version of the manuscript that addresses the points raised during the review process.

We look forward to receiving your revised manuscript.

Kind regards,

Norikatsu Miyoshi, M.D., Ph.D., FACS

Academic Editor

PLOS ONE

Journal Requirements:

Reviewers' comments:

Reviewer's Responses to Questions

**Comments to the Author**

1. If the authors have adequately addressed your comments raised in a previous round of review and you feel that this manuscript is now acceptable for publication, you may indicate that here to bypass the “Comments to the Author” section, enter your conflict of interest statement in the “Confidential to Editor” section, and submit your "Accept" recommendation.

Reviewer #3: All comments have been addressed

Reviewer #4: (No Response)

2. Is the manuscript technically sound, and do the data support the conclusions?

Reviewer #3: Partly

Reviewer #4: Yes

3. Has the statistical analysis been performed appropriately and rigorously? 

Reviewer #3: Yes

Reviewer #4: Yes

4. Have the authors made all data underlying the findings in their manuscript fully available?

Reviewer #3: Yes

Reviewer #4: Yes

5. Is the manuscript presented in an intelligible fashion and written in standard English?

Reviewer #3: Yes

Reviewer #4: Yes

6. Review Comments to the Author

Reviewer #3: (No Response)

Reviewer #4: The article is in line with the current hot topics of radiotherapy activating immunity. Using radiotherapy and single-targeted immunotherapy in combination，it is very inspiring to draw the conclusion that this combination regimen is superior to radiotherapy or immunotherapy alone. But there are still some details to discuss in this article:

1. The author's intention of using high-dose irradiation alone as part of a combined regimen is based on previous published results, from high-dose alone, or hypofractionation, or multiple fractions. The one with the most enrichment of immune cells in the tumor microenvironment after radiation has been selected. But not all radiation regimens have been considered, such as low-dose 1-2 Gy radiation，which can induce more immune cells, or flash radiation，which has been reported to have a effect of activating abscopal effect.

2.The recruitment of CD8+ cells increases after the combination therapy, and the next immune mechanism has not been explored. Currently, it is only limited to the illustration of recruited immune cells.

3. Since we all know the effectiveness of PD-1 antibody combined with radiotherapy, why do the authors choose CD47 antibody instead of PD-1 antibody? Are there specific groups of people? or specific tumor subtypes? Is there any cooperation with concurrent clinical research? No experiment on toxicity caused by immunotherapy or combination therapy is described. Even if the program improves the tumor growth rate and survival time, it is not explained whether there are toxic adverse reactions.

4. It is recommended to conduct further validation in human xenograft vectors (PDX/mini-PDX/organoids）.

7. PLOS authors have the option to publish the peer review history of their article (what does this mean?). If published, this will include your full peer review and any attached files.

Reviewer #3: No

Reviewer #4: No

---

## [Author Response · Author response to Decision Letter 1]

26 Jul 2022

We thank the editor and reviewers for their attention. We revised the article according to the reviewer's comments.

---

## [Editor Report · Decision Letter 2]

11 Aug 2022

Combining Ablative Radiotherapy and Anti CD47 Monoclonal Antibody Improves Infiltration of Immune Cells in Tumor Microenvironments

PONE-D-22-01036R2

Dear Dr. Seyed Amir Jalali,

We’re pleased to inform you that your manuscript has been judged scientifically suitable for publication and will be formally accepted for publication once it meets all outstanding technical requirements.

Kind regards,

Norikatsu Miyoshi, M.D., Ph.D., FACS

Academic Editor

PLOS ONE
---

## [Editor Report · Acceptance letter]

15 Aug 2022

PONE-D-22-01036R2 

Combining ablative radiotherapy and anti CD47 monoclonal antibody improves infiltration of immune cells in tumor microenvironments 

Dear Dr. Jalali:

I'm pleased to inform you that your manuscript has been deemed suitable for publication in PLOS ONE. Congratulations! Your manuscript is now with our production department. 

Kind regards, 

on behalf of

Dr. Norikatsu Miyoshi 

Academic Editor

PLOS ONE